# Modeling protective meningococcal antibody responses and factors influencing antibody persistence following vaccination with MenAfriVac using machine learning

Md Nasir[1], William B. Weeks[1]*, Shahrzad Gholami[1], Anthony Marfin[2], Mark Alderson[2], Troy Leader[2], Brian Taliesin[2], Rahul Dodhia[1], Juan Lavista Ferres[1], Niranjan Bhat[2]

**1** AI for Good Lab, Microsoft, Redmond, Washington, United States of America, **2** Center for Vaccine Innovation and Access, PATH, Seattle, Washington, United States of America

* wiweeks@microsoft.com

## Abstract

Meningococcal meningitis poses a significant public health burden in the meningitis belt region of sub-Saharan Africa. The introduction of the meningococcal PsA-TT vaccine (MenAfriVac®) has successfully eliminated *Neisseria meningitidis* serogroup A (NmA) cases in the region. However, the duration of post-vaccination immunity and the need for booster doses remain uncertain. To address this knowledge gap, we developed computational models using machine learning techniques to improve the effectiveness of modeling in guiding vaccination strategies for the African meningitis belt. Using serologic data from previous clinical trials of PsA-TT, we proposed a short-term and a long-term model that integrated demographic and medical variables (such as age, height and weight) with previous antibody titer levels and vaccination information to predict NmA antibody titer levels following vaccination. In the short-term model, we found moderately high performance ($R^2 = 0.59$) for out-of-training-data subjects and even better performance ($R^2 = 0.83$) in the long-term evaluation. Our models estimated the half-life of the vaccine to be 13.9 years for the study population overall, similar to previously reported estimates. Machine learning techniques offer several advantages over previous approaches, as they do not require multiple readings from the same subject, can be rigorously validated using a subset of subject data not used for training. The proposed approach also facilitates the interpretation of the relationship between input variables and antibody levels at a population level. By incorporating subject-specific demographic and medical variables, our models could potentially be used to tailor vaccination schedules to at-risk populations.

**Data availability statement:** Data cannot be shared publicly since it has sensitive information about the subjects, such as their medical history, which could be a violation of their privacy if made public. Data may be accessed by researchers from the Meningitis Vaccine Project (MVP) who created the dataset. (Contact Lionel Martellet via lmartellet@path.org)

**Funding:** The author(s) received no specific funding for this work.

**Competing interests:** The authors have declared that no competing interests exist.

## Introduction

Every year, at least 1.2 million meningitis cases are estimated to occur worldwide, resulting in about 135,000 deaths [1,2] with numbers further increased during epidemics. Epidemic Meningitis presents a major public health burden in the sub-Saharan meningitis belt [3], a region in Africa stretching from Senegal to Ethiopia.

One of the most common and severe meningitis cases are caused by infection with the bacteria *Neisseria meningitidis* (Nm). While adults get infected, children and adolescents do so at much higher rates [4]. Historically, Nm serogroup A (NmA) was responsible for 80–85% of meningococcal cases in the meningitis belt [5].

To address this need, the Meningitis Vaccine Project (MVP) and the Serum Institute of India developed MenAfriVac®, a serogroup A meningococcal polysaccharide–tetanus toxoid conjugate vaccine (PsA-TT) specifically designed for the African meningitis belt. Mass PsA-TT campaigns conducted among people 1–29 years old in the meningitis belt followed by the introduction of the vaccine into routine infant immunization in several countries has been instrumental in eliminating NmA-associated disease in the region [6,7]. Despite this public health success, there is evidence that post-vaccination immunity wanes over time [6]. Therefore an understanding of the duration of protection of PsA-TT and related meningococcal vaccines is needed to shape immunization policy, including the need for and timing of booster doses.

Long term follow-up studies that directly measured levels of serum NmA-specific bactericidal antibodies demonstrated a biphasic dynamics of the titers that declined substantially over the first 6–10 months after the initial vaccination, but this was followed by a plateau exhibiting a much slower decrease in concentrations over the next 4–5 years [8–11]. Using data from a subset of these studies, White *et al.* [12] developed a mixed effects model for analyzing and predicting the antibody dynamics of subjects following PsA-TT vaccination in Africa over even longer periods of time, producing estimates for antibody half-life that differed across age groups and the number of doses administered. While this initial work improved our understanding of the longevity of NmA vaccine-induced protection, there were a few limitations to this approach. First, the model does not incorporate subject-specific demographic and medical variables, variables that would be useful in customizing vaccination schedules for at-risk populations. Second, the model requires multiple titer readings from the same subject to make a prediction of future value. Finally, in real world long-term applications, the assumed exact exponential relationship between titers and time can lead to high compounding errors in successive predictions.

Under the current study, we developed new computational frameworks using machine learning techniques in order to improve the utility of modelling to inform decisions on PsA-TT scheduling and dosing. The current model used demographic and medical variables in combination with vaccination information to predict NmA antibody levels (as assessed by titer concentration) following vaccination. In contrast to the prior studies by White *et al.* [12], development of the current model included verification using a subset of subject data not used to train the model. In addition, the current model enables reporting of standard regression performance metrics and provides the ability to interpret the relationship between data inputs and antibody levels as the model output at a population level, thus generating an estimate of the half-life of the vaccine-induced antibody under different parameters.

## Methods

### Dataset

Demographic, physical, and serological data were obtained from four clinical trials conducted as part of the PsA-TT (MenAfriVac) development program (PsA-TT-002 [13], PsA-TT-003 [13], PsA-TT-004 [14], and PsA-TT-007 [14]) as well as follow-up serosurveys of the study populations to assess longer-term immunity (Pers-002 [11], Pers-003 [8], Pers-004 [9], and Pers-007 [10], respectively) (Table 1). These vaccination programs were conducted across clinical study sites in Senegal, Mali, Gambia, and Ghana to investigate the immunogenicity of the PsA-TT vaccine. It was noted that approximately half of participants in Pers-004 had received additional doses of multivalent meningococcal vaccines containing NmA antigen [9] due to campaigns conducted in reaction to outbreaks of other meningococcal types, while the remaining participants demonstrated a substantial degree of antibody rise over the follow-up period, possibly due to unrecalled vaccinations or cross-reactivity with commensal organisms [15–17]. Therefore, data from Pers-004 were considered inappropriate for modeling antibody decay and were excluded from subsequent analyses.

In total, data from 3130 participants enrolled in the PsA-TT clinical studies between 2006–2012 were included in the current dataset. Data for each participant included serology results from 4 to 8 samples collected during a 3 year follow-up period after the initial vaccination followed by 1 or 2 more additional samples collected during later follow-up studies occurring at least 4 years after vaccinations. By design, there was not more than one vaccination in between any two consecutive sample collections.

Variables in the dataset included demographic and clinical information (date of birth, gender, height, weight, *etc.*) and vaccination information (See supplementary information in S1 File for details). Some participants received multiple doses of PsA-TT vaccine, while some received only one dose and were administered other vaccines like Hib polysaccharide tetanus toxoid (Hib-TT), Polysaccharide serogroup-ACWY (PsACWY). For all studies, two primary serological assays were performed using the serum collected at different points during the study—rabbit complement serum bacterial antibody (rSBA) titers and *N. meningitidis* serogroup A-specific IgG enzyme-linked immunosorbent assay (ELISA) (ug/mL). Given the historical importance of the functional antibody measure to confirm the adequacy of NmA vaccine-induced responses, particularly for licensure, this analysis focused solely on rSBA titers [18].

The lower limit of quantitation (LLOQ) for the assay was 4, and values below this threshold were assigned LLOQ/2 (i.e., 2). In our dataset, less than 1% of observations were below the LLOQ.

**Table 1. Overview of datasets used.**

| Name of the study (Subjects enrolled initially, Total N = 3130) | Participant age at the time of enrollment | Study Site Locations | Number of serum collections by time interval after study entry[1] (Number of subjects with continued participation) | | | |
|---|---|---|---|---|---|---|
| | | | PsA-TT serum collections | | | Persistence (Pers) serum collection* |
| | | | Within 1 month | 1 month-1 year | 1-3 years | After 4 years or later |
| PsA-TT-002 (330) | 12-23 months | Bamako, Mali; Basse, The Gambia | 2 | 3 | 2 | 1 |
| PsA-TT-003 (600) | 2-29 years | Bamako, Mali; Basse, The Gambia; Niakhar, Senegal | 2 | 2 | – | 1 |
| PsA-TT-004 (1000) | 14-18 weeks | Navrongo, Ghana | 2 | 1 (variable) | 1 (variable) | N/A |
| PsA-TT-007 (1200) | 9-12 months | Bamako, Mali | 2 | 2 | – | 1-3 |

*Collected under a separate study protocol (Persistence-002,003, and 007) and 4–5-year time point post-vaccination. Pers-004 was not included in current work.

The original clinical trials from which these data were generated was approved by multiple ethics committees and boards: London School of Hygiene and Tropical Medicine, Centre for Vaccine Development (Bamako, Mali), Navrongo Health Research Centre Institutional Review Board - Ghana Health Service (Navrongo, Ghana), and Institut de recherche pour le developpement-Senegal (Dakar, Senegal) and individual informed consent was obtained during their conduct. The current secondary retrospective analysis was reviewed and approved with waiver of re-consent by the same ethical committees. The data was fully anonymized to ensure no personally identifiable information on the participants. In the scope of the current study, the dataset was accessed from May 24, 2021 to February 13, 2024 for secondary retrospective analysis.

### Inclusivity in global research

Additional information regarding the ethical, cultural, and scientific considerations specific to inclusivity in global research is included in the Supporting Information in S2 File.

### Modeling

The composite nature of the current dataset resulted in substantial variability in the number of measurements for each subject, the number of vaccinations, and the time elapsed between the events of vaccination. Due to the absence of temporal alignment among the samples and small number of titer reading instances per subject, we chose not to apply time series-based models. Instead, we decomposed the modeling of the antibody kinetics into independent tasks of predicting future antibody levels from a past measurement along with other variables based on vaccination history and demographic information. To achieve this, we used regression techniques which sought to predict the antibody level (as represented by rSBA titer and denoted as $Ab_n$) at a given point of time $t = t_n$. In this setup, each measurement – with the exception of the initial reading, which had no previous information – from each subject was considered one independent sample (target variable) in the data that we used for training and testing the predictive model.

We proposed two alternative modeling approaches based on which prior antibody level is used to predict the future level:

1. Long-term: In this model we used the baseline antibody level $Ab_0$ recorded at the beginning of the study (indicated by time $t = t_0$) typically measured from a specimen collected shortly before the first vaccination (illustrated in Fig 1a). The time elapsed between the two readings ($t = t_n - t_0$ et al. [12] and formally elaborated in the supplementary information in S1 File. Next, we derived several features (covariates) from the vaccination information during this time period (ranging from one to four vaccinations), including the timing, number, type, and dose of the vaccination. To obtain standardized features from the timing information in the case of multiple vaccine doses, we used the time of the first and the last vaccination (following the initial dose) and the mean of the time interval between all doses. Finally, we included subject-specific demographic information in the model. In potential real-world applications, this model would be suitable for applying directly to simulate and test and would need to be run only once with the baseline level as input to estimate the antibody level at any target time. A model limitation is that the estimate could be less accurate for small values of $t$ as the model is unlikely to encounter such time ranges with much frequency.

2. Short-term: Prior studies have shown that the antibody level has an exponential relationship with time [12]. As a result, limiting the training samples to shorter ranges of time makes the model less prone to error, particularly if multiple readings from the same subject are available. We therefore proposed an alternate short-term modeling approach which simplified the prediction task by looking ahead only one step at a time. More specifically, to predict $Ab_n$, we used the last available reading $Ab_{n-1}$ (taken at time $t_{n-1}$) instead of $Ab_0$ (illustrated in Fig 1b). In this model, by design, only one or no vaccine dose was administered during this period between measurements. When given multiple true readings of

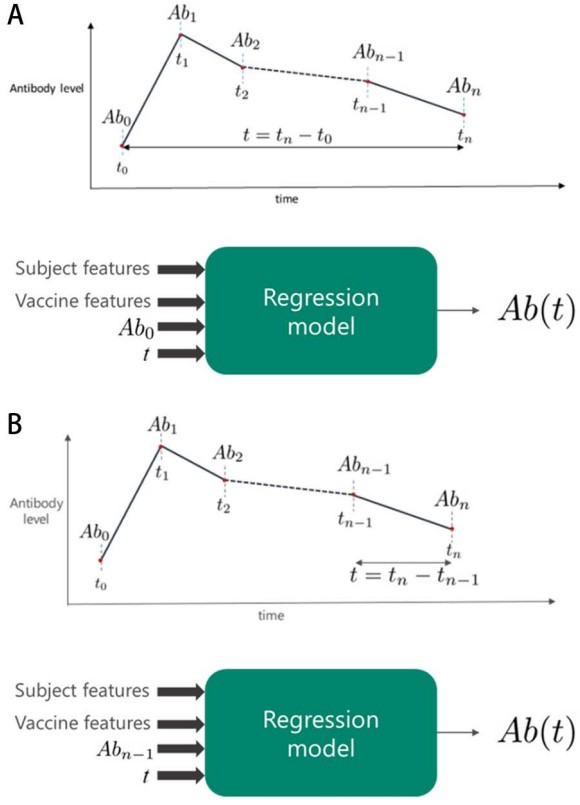

**Fig 1. a-b. Schematic representation of the long-term and short-term modeling, respectively.** a) Long-term modeling approach includes the baseline antibody level $Ab_0$ only and no further readings, b) Short-term modeling approach includes the baseline antibody level $Ab_{n-1}$ only and no prior readings.

the antibody level, the model uses a smaller value of $t$ and, hence, is less prone to the aforementioned potential error due to the exponential relationship of the time and antibody level. However, if applied for a new study with only the baseline reading available, the short-term model might have compounding errors to predict antibody levels at multiple points of time in the future while using the predicted level at each point as baseline for the next prediction. To be accurate, this approach requires tracking the antibody trajectory via stepwise serial iterations using the predicted antibody level as the input to predict the next antibody level. It is worth noting that the impact of the past vaccination history (before $t_{n-1}$) is implicitly used by the model via the last antibody level $Ab_{n-1}$, avoiding the need to explicitly include them (See supplementary information in S1 File for more details).

## Regression

We obtained several variables to associate with a past and future antibody level pair, which constituted one data sample for the regression task. From our combined dataset, we obtained 13750 such samples, with each subject typically having multiple reading pairs. Dropping missing values resulted in 11776 samples for the long-term model and 11771 samples for the short-term one. Due to variations in the design of the original constituent studies—such as differences in the number of vaccinations and the timing of vaccine administration—we derived summary variables for the long-term model. For instance, we used the mean time of vaccination rather than individual vaccination time points. The long-term model had

13 input variables in total; the short-term model had 11 variables. After grouping samples from the same subject to eliminate data contamination, we randomly split the dataset into train (80%) and test (20%) subsets.

We used the state-of-the-art interpretable regression model called Explainable Boosting Machine (EBM) [19]. EBM is an explainable supervised predictor model based on modern machine learning techniques such as bagging and gradient boosting. In addition to using main features, the model incorporates pairwise interaction effects between them to represent different combinations. The antibody level to be predicted was modeled as a linear combination of the main and interaction features using a generalized Gaussian Additive Model (GAM).

For comparison, we also used a number of widely used regression models as baselines: linear regression, regression tree, LightGBM [20], and XGBoost [21]. Linear Regression and Regression Trees are inherently interpretable and serve as useful baselines. LightGBM and XGBoost, while more complex, provide feature importance metrics and can be interpreted. Although we considered other ensemble and deep learning methods, we excluded them due to their limited interpretability in the context of this study's objectives.

For linear regression, the predictor variables were normalized, and the categorical variables were encoded with one-hot encoding. EBM and the other three baseline models did not require any such preprocessing due to their scale-agnostic nature. To assess regression performance, we used two metrics: R-squared (the higher the better) and Root Mean Squared Error (RMSE; the lower the better). While these metrics share a monotonic relationship and favor the same models by design, we included both—RMSE to represent the errors using the scale of the actual values of antibody levels and R-squared to provide a standardized scale agnostic goodness-of-fit.

We conducted a 5-fold cross validation over the training data to obtain the optimal parameters for EBM and used them for finally training the model on the entire training set. Note that the logarithm of the antibody titer level is used for all reporting.

In addition to the random 80–20 data split of the entire dataset, we conducted an experiment to evaluate the model performance on 20% of the Pers-007 dataset only as the test split to better simulate the real-world application scenario wherein long-term protection provided by the vaccine could be assessed. These data were derived from a study population considered to have a low risk of external exposures to NmA antigens following their initial vaccination course. For this experiment, we only used EBM as the regressor and different combinations of training data while ensuring there was no overlap of subjects between training and test splits.

### Model interpretation

To interpret how the best performing regression systems (short- and long-term models with EBM) made predictions, we analyzed the importance of features on the predicted target antibody level. Due to the additive nature of EBM, the weights assigned by the trained model to different features (and some of their pairwise interactions) are indicative of their relative contribution.

**Analysis of the impact of different features on antibody level.** We analyzed how variations in different feature values contributed to the antibody level estimation based on the trained EBM regression model. Each of these features was represented by a distribution as seen in the training data and the model learned to assign a level of contribution towards the output (logarithm of the rSBA titer) for different values in that distribution. The interpretable "glassbox" (transparent) nature of our model allowed for analysis of the impact of any feature used in the framework. In the analysis of the contribution of each individual feature, the "score" describes how much value is added to the predicted output (logarithm of rSBA titer reading) for a specific feature if other features remain unchanged. The "density" plot shows the distribution of values of the corresponding feature. However, given the aims of the current study, we chose to focus on the following three features: (i) which vaccine(s) the subject was given, (ii) the time elapsed since the first dose of PsA-TT, and (iii) the gap between two doses of PsA-TT vaccine. For the analysis of (ii), we used samples with exactly one dose of PsA-TT vaccine; for the analysis of (iii), we used samples with exactly two PsA-TT doses.

## Half-life estimation

We defined the population half-life of the vaccine to be the time elapsed from vaccination to the point when half of the vaccinated population had rSBA antibody titers above 128, which is considered a protective antibody level in humans. This threshold is supported by prior validation studies showing that rSBA titers ≥128 reliably predict protection against meningococcal disease [22]. To estimate the population half-life, we applied the trained long-term model on all data instances prior to receipt of a second vaccination (if any) for individuals whose first vaccination was MenAfriVac. For those data instances, we used the initial antibody level as the input to the model ($Ab_0$) if it was above the threshold, otherwise we used the highest antibody level for the individual. It should be noted that our definition of half-life is conditioned on a positive response to the vaccine. Hence if an individual had no recorded antibody levels above the threshold, the subject was excluded from the analysis. To generate 95% confidence intervals, we used a bootstrapping methodology wherein we randomly sampled 1000 subjects from the selected pool without replacement and predicted their future antibody levels one day at a time, incrementally applying the model for the next day using the previous day's antibody level as an input. For each subject, we calculated the time between vaccination and when rSBA first fell below 128; the median calculated time for the 1000 subjects was the half-life for that sample. We repeated the experiment 30 times by selecting different samples of 1000 subjects and calculated the mean and 95% confidence intervals of half-life. Next, we performed the same procedure for two subgroups in the data based on age: subjects aged under 2 years and over 2 years.

## Results

### Regression performance

The R-square and RMSE values for both long- and short-term models are shown in Table 2. The EBM model performed better than all other comparator regression models, both in long-term and short-term approaches, as reflected in the reported metrics..

Fig 2 shows the distribution of residual errors of the two EBM models: when compared to the long-term model, the short-term model largely had smaller residuals that were centered around zero and were visually closer to a normal distribution, indicating its superior performance. (Fig 2b).

For longitudinal evaluation, the long-term model performed better than the short-term model in all training dataset scenarios (Table 3). Using validation samples from the same dataset as the test data achieved the best performance, which was expected as all such samples were collected under the same study protocol. Finally, we found that using more samples resulted in the best performance despite possible variability between training and test data, suggesting that more data might further improve our models in the future, irrespective of the study protocol.

Fig 3 shows reference and predicted trajectories for a few sample subjects. We observe that our model can closely track the dynamics of the antibody levels for different vaccination groups.

Table 2. Performance of different models and regression methods on test data. R-squared and RMSE metrics are used.

| Regression method | R-squared | | RMSE | |
|---|---|---|---|---|
| | Long-term model | Short-term model | Long-term model | Short-term model |
| Linear Regression | 0.20 | 0.15 | 2.33 | 2.33 |
| Regression Trees | 0.38 | 0.35 | 2.05 | 2.05 |
| LightGBM | 0.45 | 0.43 | 1.93 | 1.91 |
| XGBoost | 0.46 | 0.58 | 1.89 | 1.66 |
| EBM | **0.52** | **0.59** | **1.81** | **1.63** |

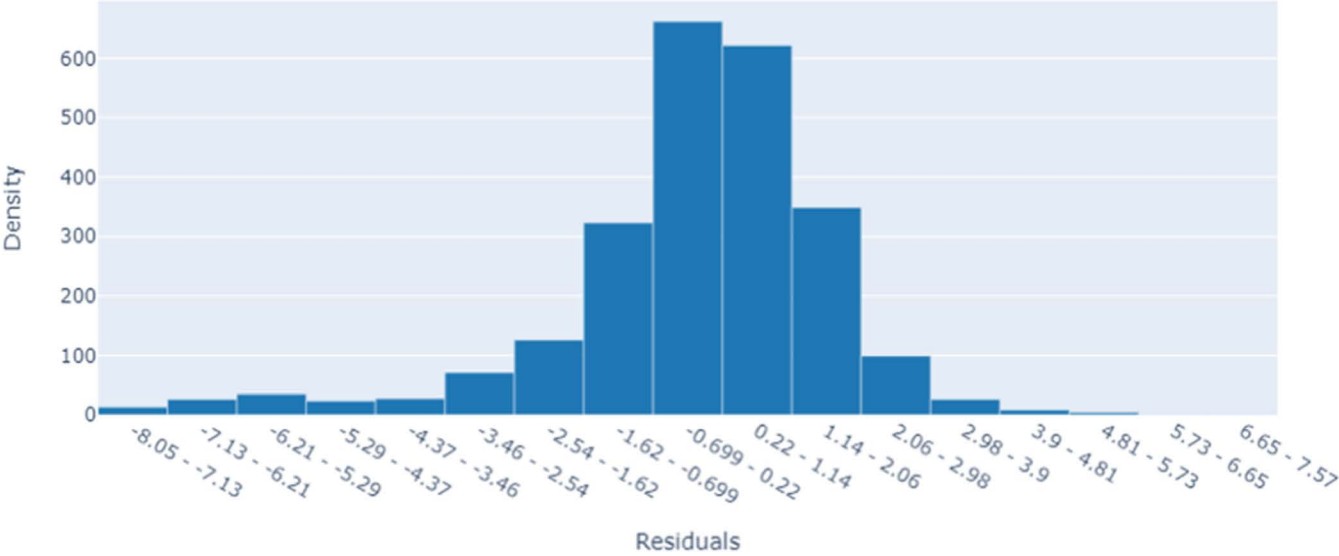

A   RMSE = 1.81 | R² = 0.52

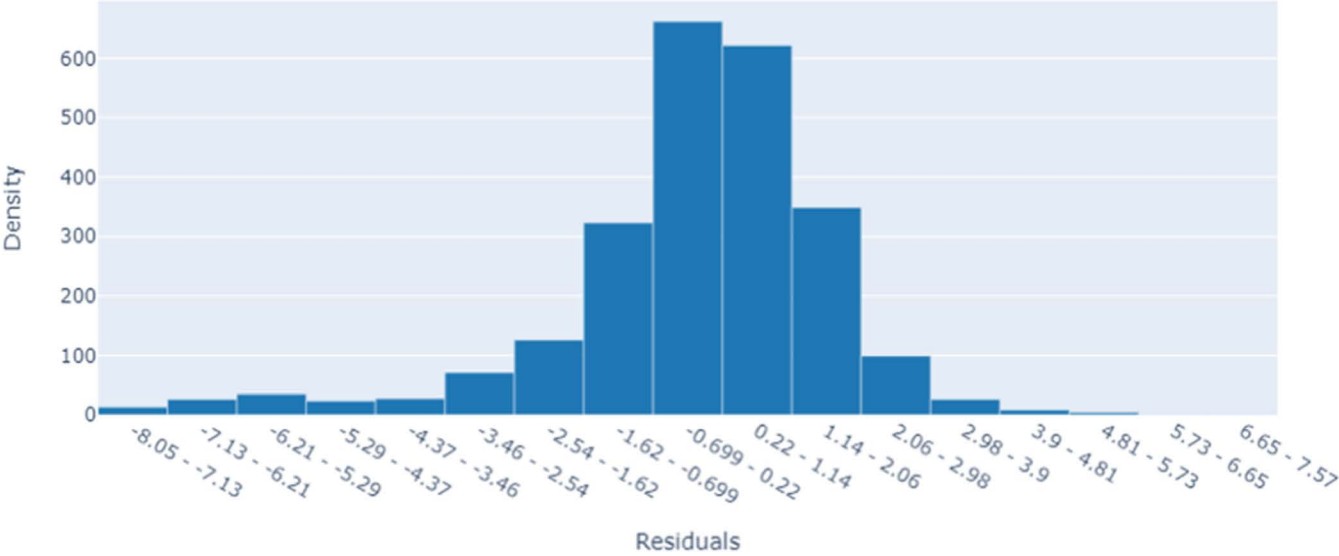

B   RMSE = 1.63 | R² = 0.59

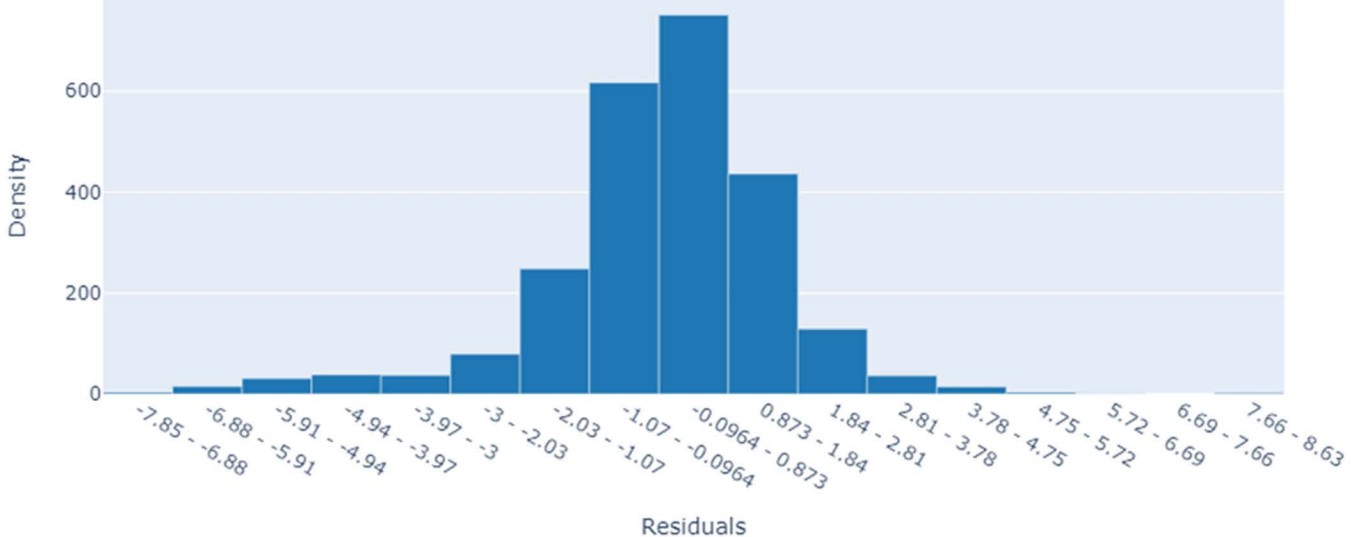

**Fig 2.  a-b: Distribution of errors (residuals) on test set with long-term model and EBM.** a) Long-term model, b) Short-term model.

## Model interpretation

Fig 4a and 4b show the relative importance of the most influential features (and their combinations or interaction terms) in prediction, as learned by the model. Both models were heavily dependent on the time between the two readings. Some other influential features common to both models were the last antibody level, and the type and the dose of the vaccines. However, there were considerable differences in the list of features between these two models, due to inherent differences in their design and feature engineering.

**Table 3. R-squared in longitudinal performance evaluation of the models with 20% of Pers-007 as test data for different training datasets. Subjects under the test set were excluded from each training set.**

| Training set | Long-term | Short-term |
|---|---|---|
| 80% Pers-007 | 0.58 | 0.40 |
| Pers-002 + Pers-003 + 80% Pers-007 | 0.70 | 0.42 |
| All PsATT + Pers-002 + Pers-003 + 80% Pers-007 | **0.83** | **0.45** |
| All PsATT + Pers-002 + Pers-003 | 0.30 | 0.22 |

**Feature analyses.** i) **Impact of different vaccine groups:** The impact of different vaccine groups is shown in **Table 4**. We found that one or two doses of PsA-TT were most effective in eliciting antibody responses. Hib-TT or no vaccine is associated with decline in immunity, reflecting the absence of antigen-specific re-stimulation. Similarly, when PsA-TT and Hib-TT were given in sequence, the order of administration had minimal effect. As expected, the combination of the conjugate vaccine PsA-TT followed by the polysaccharide-only vaccine PsACWY improved responses compared to a single dose of PsA-TT.

ii) **Impact of time since first vaccination:** Fig 5 shows the impact of time elapsed since the first dose of PsA-TT, in the absence of any more subsequent dose. As expected, antibody levels decreased as more time elapsed.

iii) **Impact of time gap between two vaccine doses:** When analyzing samples with exactly two doses of PsA-TT, we did not observe a meaningful relationship between the time gap between doses and the resulting antibody level. However, most of the time intervals were concentrated between 244 days and 285 days, making it infeasible to obtain a reliable estimate of the association. We did observe that a longer gap between vaccinations might be associated with higher immunity, however this trend lacks support beyond the aforementioned range due to insufficient sample size.

## Half-life estimation

Using a predefined rSBA threshold of 128, we estimated the mean vaccine half-life for the given population to be 13.9 years (95% confidence interval [CI]: 3.1 years to 23.0 years), slightly lower than but close to the estimated 16.5 years previously reported by White et al. [12]. Subjects under 2 years of age at the time of vaccinations had an estimated half-life of 12.4 years (95% CI: 4.2, 22.0) while older subjects (2 years or older) were estimated to have slightly longer half-life, 15.8 years (95% CI: 6.1, 21.5). The estimated half-life is higher than most other vaccines, for which immunity wanes after 3–5 years, though this duration varies by serotype and by the protein used for conjugation [23,24].

## Discussion

The MenA conjugate vaccine and the herd immunity it conferred have nearly eliminated meningococcal A meningitis in sub-Saharan Africa. Consequently, in the absence of re-introduction of the pathogen and the return of outbreaks, it is difficult to evaluate the persistence of serum bactericidal antibody titers as a surrogate of protection. Studies directly measuring the presence of antibody have shown that titers persist above a putative protective threshold for up to 5 years [25]. However, additional efforts to follow up the same participants from these studies for longer periods are operationally prohibitive. Therefore, modeling antibody kinetics is the most practical method for predicting the duration of protection. In this article, we describe the formulation and evaluation of a machine learning-based framework to characterize and predict the kinetics of serum antibody response to meningococcal serogroup A vaccination. Using this novel approach, we were able to confirm and quantify key influential modifying factors in the longevity of the immune response and generate plausible estimates of duration of protection.

Traditional statistical regression approaches have provided initial predictions that appear reasonable but with wide ranges of uncertainty, estimating the half-life of the antibody response to be 7.4 years (95% CI 2.8–41.3) for young

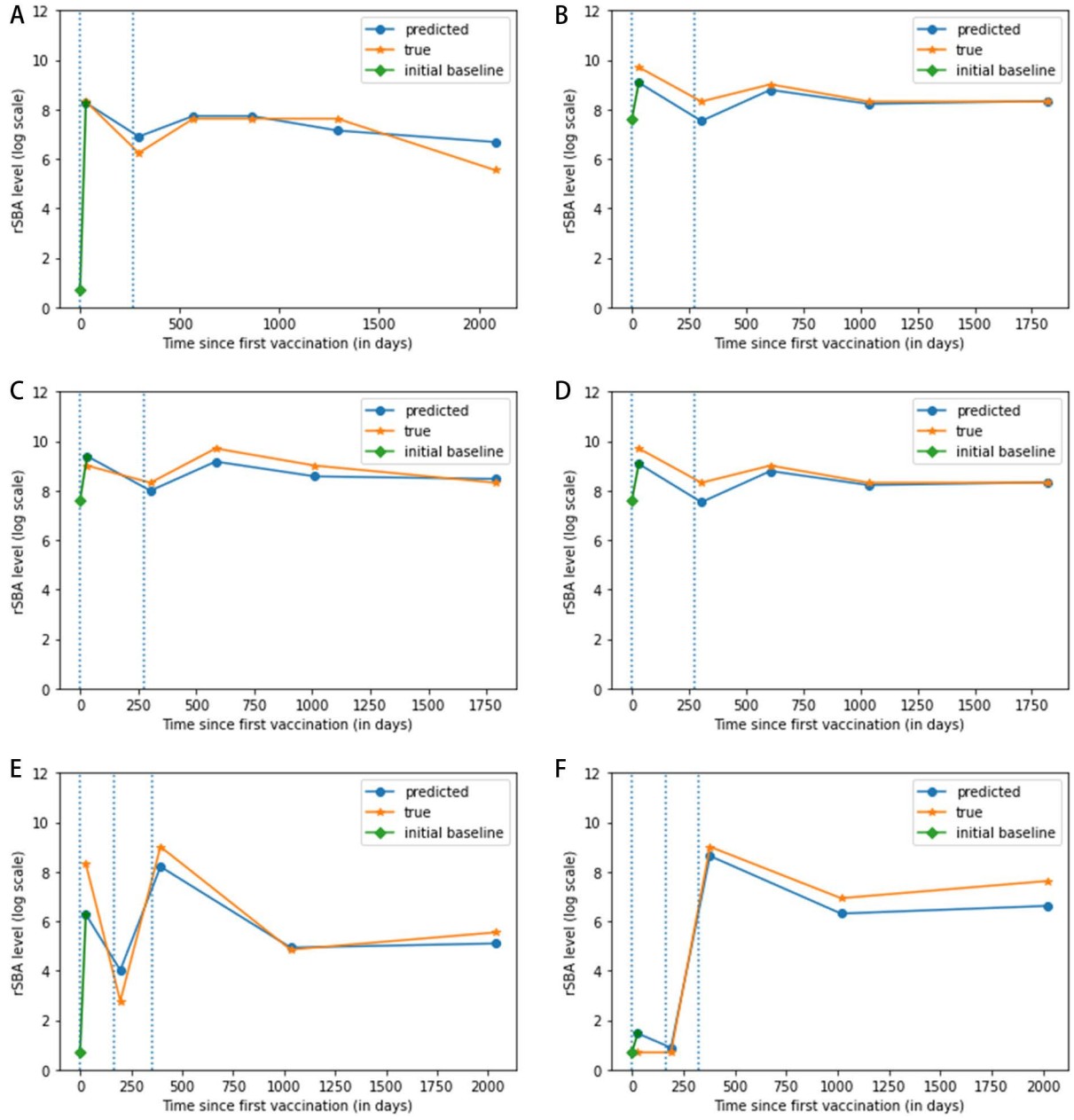

**Fig 3. a-k: Sample trajectories of the predicted and true reference rSBA titer levels for different individual subjects.** The vertical lines represent the vaccination times and the stand-alone (diamond-shaped) points represent the initial baseline for the rSBA titer levels. The type of the vaccines given (no vaccine or a placebo was given in some cases, indicated by 'none') is mentioned under each subfigure. a) 1st: PsA-TT, 2nd: Hib-TT, b) 1st: PsA-TT, 2nd: 1/5th dose of PsACWY, c) 1st: PsA-TT, 2nd: 1/5th dose of PsACWY, d) 1st: PsA-TT, 2nd: none, 3rd: PsA-TT, e) 1st: none, 2nd: none, 3rd: PsA-TT, f) 1st: none, 2nd: none, 3rd: PsA-TT, g) 1st: none, 2nd: none, 3rd: PsA-TT, h) 1st: PsA-TT, 2nd: Hib-TT, 3rd: PsA-TT, i) 1st: PsA-TT, 2nd: Hib-TT, 3rd: PsA-TT, j) 1st: PsA-TT, 2nd: Hib-TT, 3rd: PsA-TT, k) 1st: PsA-TT, 2nd: Hib-TT, 3rd: PsA-TT.

children and 16.5 years (95% CI 7.7–39.1) for the 2–29 year age group [12]. Our approach expanded on this prior work by using a novel machine learning methodology that has not typically been used in public health research settings. The advantage of this approach is that instead of using a single mixed-effect model that follows an assumed biphasic exponential decay function, we used two models, each applied to two time ranges reflective of the two phases of antibody

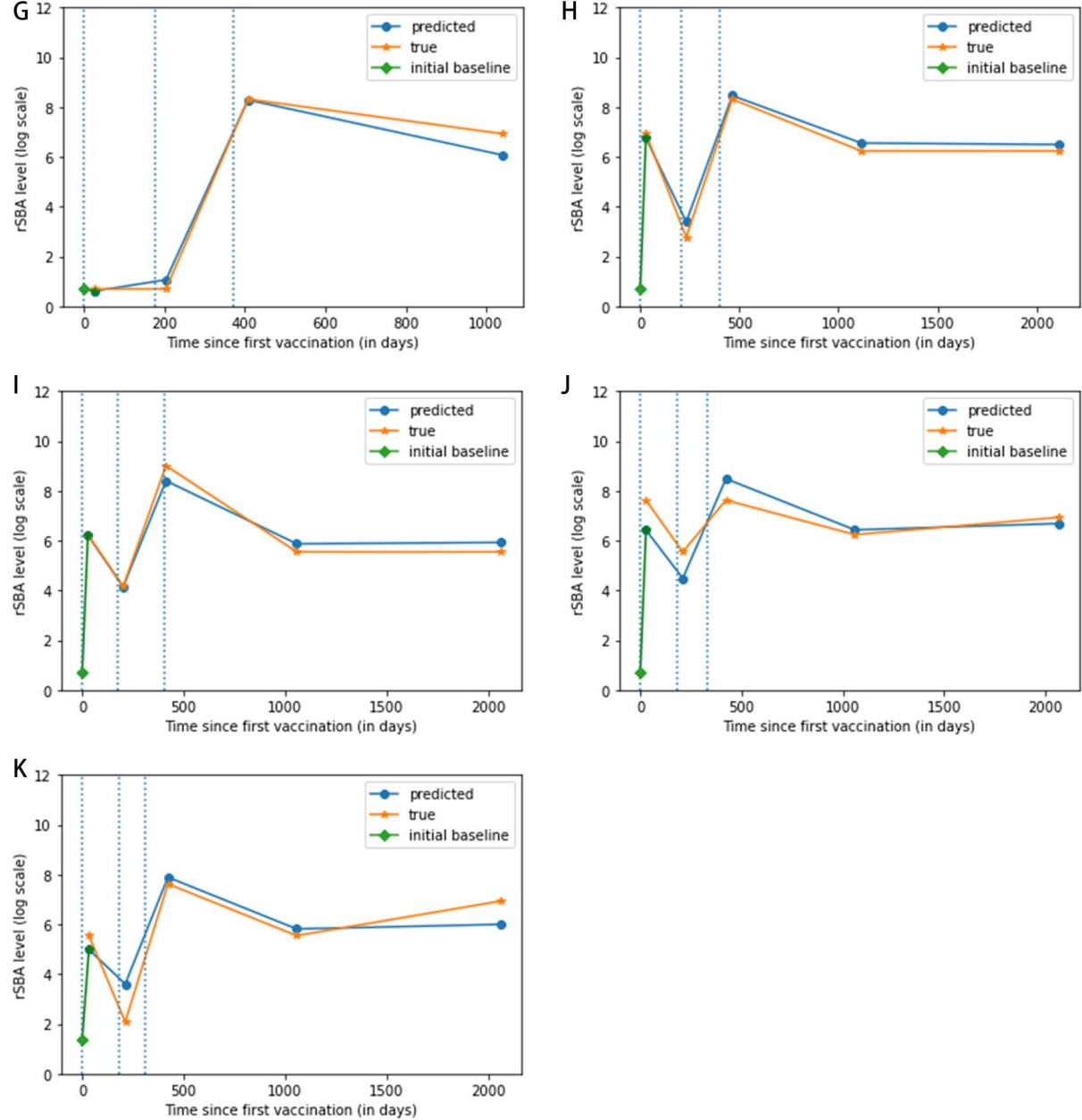

**Fig 3.** Continued.

decay. Our approach did not rely on prior knowledge of temporal dependence between sequential measurements in the same individual; instead, the model learns temporal dependence from the available data, making it arguably more suitable for real world heterogeneous datasets of varying qualities. In addition, while our model used multiple data points from the same subject during training, we were able to validate the model using data from a separate group of randomly selected subjects, thereby avoiding overfitting. This subject-agnostic training approach allows the model to estimate future antibody levels of any new subject who has at least one antibody measurement, in contrast to mixed effect models which require multiple readings from a subject to first estimate the subject-specific parameters before any prediction is made.

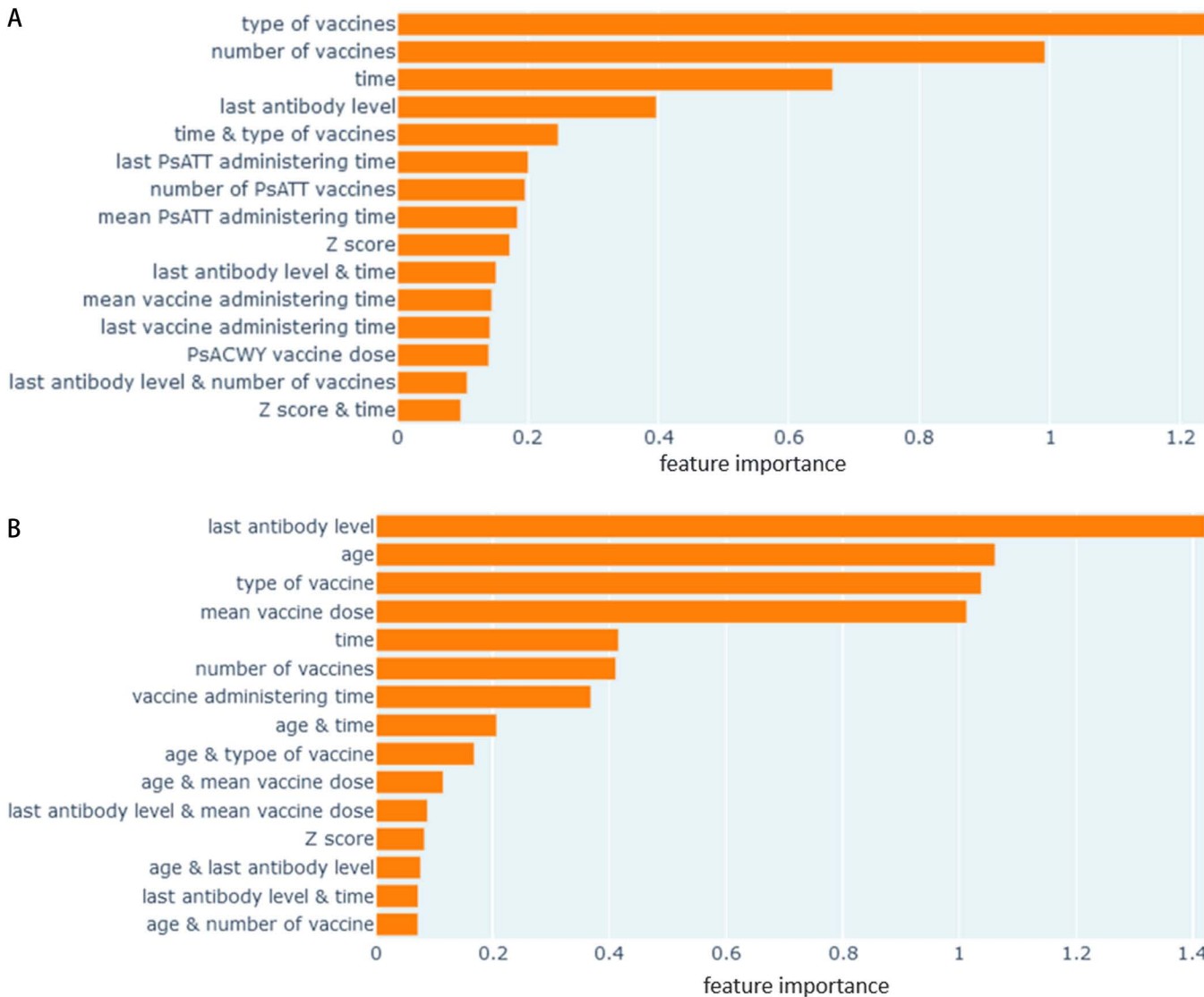

**Fig 4. a-b: Relative importance (as weights) of top 15 features for long-term and short-term model with EBM.** a) Long-term model, b) Short-term model.

Our approach of using EBM as a regression algorithm also provides a quantitative interpretation of the prediction process by identifying different meaningful features (such as age or number of doses) that were deemed useful by the model and their relative degree of influence in the prediction task. The results also validated prior findings [26] that subject *age* at initial vaccination is an important predictor. The prior antibody level was more important for the short-term model than for the long-term model. The long-term model's most influential feature was the number of PsA-TT doses administered, a variable not available to the short-term model. While these relative importance measures are useful indicators for understanding the model's mechanism, they could still be affected by confounding variables that are not part of the dataset.

With EBM, the short-term model performed better in terms of prediction metrics (both RMSE and R-squared) than its long-term counterpart. However, this better performance was not seen when the alternate reference algorithms were used for regression. This could be attributed to the additional interaction terms incorporated in EBM which might have helped

**Table 4. Impact of different vaccine groups on immunity. Groups with less than 100 subjects are excluded.**

| Vaccine Group | | Impact weight | Number of Samples |
|---|---|---|---|
| PsA-TT only | PsA-TT (single dose) | 0.581 | 5542 |
| | PsA-TT (two doses) | 0.798 | 2327 |
| combined | PsA-TT + Hib-TT | −0.564 | 256 |
| | Hib-TT + PsA-TT | −0.238 | 219 |
| | PsA-TT + PsACWY | 0.453 | 248 |
| | PsACWY+PsA-TT | 0.619 | 239 |
| Non-PsA-TT only | PsACWY only | −1.468 | 128 |
| | Hib-TT only | −2.267 | 118 |
| No vaccine | | -0.104 | 1637 |

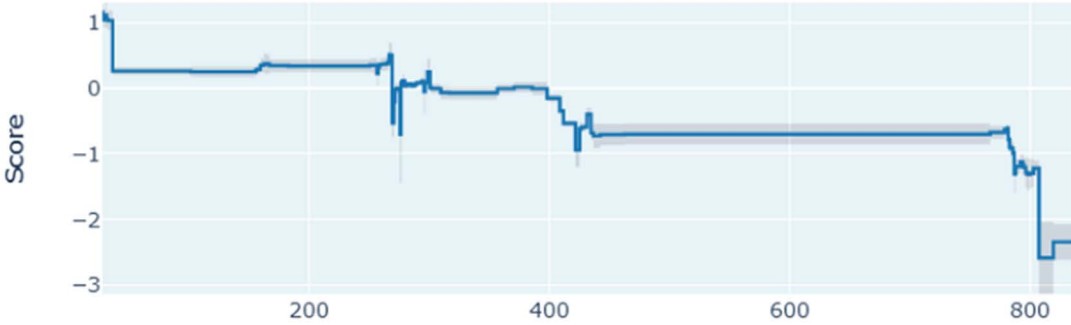

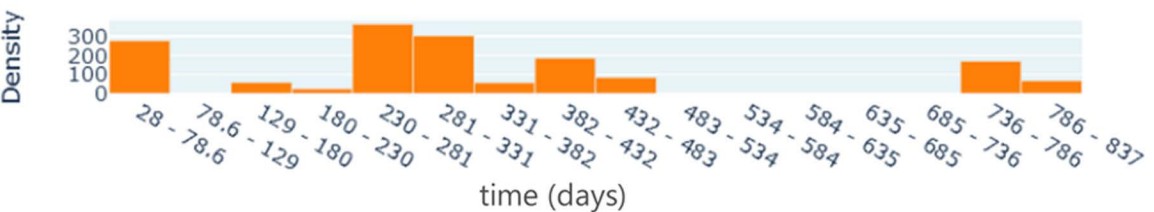

**Fig 5. Impact of the time\* elapsed since first vaccination.** \*We should note that certain ranges (78.6–129 and 483–736 days) did not have enough sample to obtain a reliable estimate of the impact, while the drop in immunity around 280 days, 430 days and 800 days were obvious as there are more samples around these timepoints.

improve the performance of the short-term model (which has fewer features in absence of interaction) over the long-term counterpart.

The modeling approach proposed in this work is an initial attempt toward using machine learning for prediction and analysis of immune response to meningococcal vaccination. There are several ways that the model might be improved with future iterations. For example, while this approach is better able to handle less structured data, obtaining large data-sets collected in a more consistent way could improve the accuracy of the model even further. Antibody level readings at more time points with regular intervals could also be useful to train a more temporally informed model capable of more accurate and detailed analysis of the immune response over time. Future work for modeling also includes accounting for the confounding variables and explicitly hard-coding biological constraints such as the exponential relationship of the immune response with time.

While the deployment of PsA-TT through campaigns and routine immunization over the past several years has been tremendously successful in eliminating NmA disease, outbreaks and sporadic cases of meningococcal disease due to other serogroups have continued to occur throughout the African meningitis belt, and are currently managed through reactive campaigns that use higher-cost multivalent conjugate vaccines available through an international stockpile [27]. Recently, a lower-cost pentavalent conjugate vaccine developed specifically for the African region was prequalified [28,29] by the WHO, and is intended for routine use in countries of the meningitis belt. As policy recommendations related to this new vaccine are developed, many of the same questions regarding duration of immunity and the need for booster immunization will be pertinent. Notably, this latter formulation is manufactured using the similar technology as PsA-TT, and there is evidence to believe that immune responses to this new vaccine will behave somewhat similarly. Therefore, as the predictive approaches explored in this report are refined, they have the potential to contribute to the evaluation and implementation of this and future meningococcal vaccines.

## Conclusion and future work

In this work, we propose two alternate modeling approaches for predicting antibody kinetics in subjects receiving meningococcal A vaccination. We find that generally an approach that models responses following a single dose works better than one that models the outcome of multiple doses in terms of regression performance, while EBM, and approach that uses machine learning, is shown to be the best regression algorithm. Given the limited number of censored values, we did not apply censored regression techniques. However, future studies with a higher fraction of censored values may benefit from such approaches to account for measurement limitations more rigorously.

## Supporting information

**S1 File. Supplementary information.**
(DOCX)

**S2 File. Checklist (Inclusivity in global research).**
(DOCX)

## Acknowledgments

We thank Yuxiao Tang and Yixi Xu for valuable discussions and feedback on this work. The data for this analysis was obtained following a request to Meningitis Vaccine Project's data sharing scheme for which we thank the PATH and the Serum Institute of India. We thank all study participants for contributing to primary data that was used for the analysis in this publication. Finally, we also thank the local investigators for their contributions in the data collection process: Samba Sow (CVD-Mali, Bamako, Mali), Olubukola Idoko (LSHTM, MRC-Gambia), Aldiouma Diallo (Institut de recherche pour le developpement-Senegal, Dakar, Senegal), and Patrick Ansah (Navrongo Health Research Centre, Ghana Health Service, Navrongo, Ghana).

## Author contributions

**Conceptualization:** William B. Weeks, Shahrzad Gholami, Anthony Marfin, Mark Alderson, Juan Lavista Ferres, Niranjan Bhat.

**Data curation:** Anthony Marfin, Mark Alderson, Troy Leader, Niranjan Bhat.

**Formal analysis:** Md Nasir.

**Investigation:** Md Nasir, Niranjan Bhat.

**Methodology:** Md Nasir, Shahrzad Gholami.

**Project administration:** William B. Weeks, Troy Leader, Brian Taliesin, Rahul Dodhia.

**Software:** Md Nasir.

**Supervision:** William B. Weeks, Rahul Dodhia, Juan Lavista Ferres, Niranjan Bhat.

**Validation:** Md Nasir.

**Visualization:** Md Nasir.

**Writing – original draft:** Md Nasir, Niranjan Bhat.

**Writing – review & editing:** Md Nasir, William B. Weeks, Shahrzad Gholami, Anthony Marfin, Mark Alderson, Troy Leader, Brian Taliesin, Rahul Dodhia, Juan Lavista Ferres, Niranjan Bhat.

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
