## [Decision Letter · Decision Letter 0]

5 Aug 2024

PONE-D-24-05900Modeling protective meningococcal antibody responses and factors influencing antibody persistence following vaccination with MenAfriVac® using machine learningPLOS ONE

Dear Dr. Nasir,

Thank you for submitting your manuscript to PLOS ONE. After careful consideration, we feel that it has merit but does not fully meet PLOS ONE’s publication criteria as it currently stands. Therefore, we invite you to submit a revised version of the manuscript that addresses the points raised during the review process.

We look forward to receiving your revised manuscript.

Kind regards,

Oyelola A. Adegboye, PhD

Academic Editor

PLOS ONE

4. Please include a complete copy of PLOS’ questionnaire on inclusivity in global research in your revised manuscript. Our policy for research in this area aims to improve transparency in the reporting of research performed outside of researchers’ own country or community. The policy applies to researchers who have travelled to a different country to conduct research, research with Indigenous populations or their lands, and research on cultural artefacts. The questionnaire can also be requested at the journal’s discretion for any other submissions, even if these conditions are not met.  Please find more information on the policy and a link to download a blank copy of the questionnaire here: https://journals.plos.org/plosone/s/best-practices-in-research-reporting. Please upload a completed version of your questionnaire as Supporting Information when you resubmit your manuscript.

5. In the online submission form, you indicated that your data is available only on request from a third party. Please note that your Data Availability Statement is currently missing the name of the third party contact or institution / contact details for the third party, such as an email address or a link to where data requests can be made. Please update your statement with the missing information.

Reviewers' comments:

Reviewer's Responses to Questions

**Comments to the Author**

1. Is the manuscript technically sound, and do the data support the conclusions?

Reviewer #1: Partly

Reviewer #2: Yes

2. Has the statistical analysis been performed appropriately and rigorously? 

Reviewer #1: I Don't Know

Reviewer #2: I Don't Know

3. Have the authors made all data underlying the findings in their manuscript fully available?

Reviewer #1: No

Reviewer #2: No

4. Is the manuscript presented in an intelligible fashion and written in standard English?

Reviewer #1: Yes

Reviewer #2: Yes

5. Review Comments to the Author

Reviewer #1: The manuscript “Modeling protective meningococcal antibody responses and factors influencing antibody persistence following vaccination with MenAfriVac® using machine learning” by Nasir et al. addresses the public health challenge posed by meningococcal meningitis in the meningitis belt region in Sub-Saharan Africa. The authors argue that despite the introduction of the MenAfriVac vaccine to eliminate cases of Neisseria meningitidis serogroup A (NmA) in this region, the duration of post-vaccination immunity and the necessity of booster doses remains unclear. To address this gap, the authors propose a long and short-term model that uses demographic, medical and antibody variables to predict NmA antibody responses, as well as vaccine’s half-life. They use Explainable Boosting Machines (EBM) to model the long and short-term models and later they benchmark these results with linear regression and decision trees, where the EBM model shows better performance.

The study is well motivated by the necessity of assessing post-vaccination immunity. It also proposes two interesting approaches (long and short-term) for predicting antibody kinetics. However, the manuscript has in my opinion some shortcomings:

1. Major issues

1.1. Introduction

- The authors state the following: “Under the current study, we developed new computational models using machine learning techniques in order to improve the utility of modeling to inform decisions on PsA-TT scheduling and dosing”. It is unclear to me whether they refer to the approach/formulation of long and short-term models or rather to the EBM model. If they refer to the former, this is not really a new development of a computational model but a rather new formulation on how to predict antibody kinetics based on long and short term responses. If the authors refer to the latter, it should be stated more clearly that they have contributed to the development of the EBM tool.

1.2. Dataset

- The exact variables which are available are unclear. Some demographic information is mentioned in the text and the authors refer to the supplementary information for the reader to check information on vaccination data. However, no such information was found in the supplementary S1. Referring to other variables as “etc” in the main text is not sufficient if an explicit list is not available in the supplementary. Table S1 contains information on variables incorporated into the long and short-term models, but as demographic information is missing from that table, it is unclear which data is really available for the authors.

-  There is no description of the cohort on the observables from the data collected (i.e., distribution of demographic variables, vaccination doses, clinical data). Descriptive statistics of the study variables will benefit the understanding of the cohort’s characteristics.

1.3. Modelling

General

-  The formal equations of the model in the supplementary material S1 are only referred to at the end of the short-term model description in the main text. Having the reference earlier (i.e., when describing the long-term model) could enhance clarity of the model formulation.

-  The authors model each antibody task independently. This might lead to the loss of shared information between the tasks.

- Antibody information often comes with a lower/upper threshold of the data. Is this the case here? If yes, the authors should consider censored regression using appropriate loss functions.

Long-term modeling: 

- While figure 1 a-b illustrates well which antibody readings are used for the prediction task, the readability of the text could benefit from including the information that the model only uses the baseline antibody level and no further readings are included.

- The authors claim that “A model limitation is that the estimate could be less accurate for small values of as the model is unlikely to encounter such time ranges with much frequency.” This should follow from the aforementioned sample description with descriptive statistics.

Short-term modeling:

- The authors state that “Prior studies have shown that the antibody level has an exponential relationship with time.[11] As a result, limiting the training samples to shorter ranges of time makes the model less prone to error, particularly if multiple readings from the same subject are available.” I agree that shorter time-intervals might reduce the variance of a prediction task but I do not see why this should be, relatively, higher for exponential relationships if the applied model allows for non-linear relationships.

- The authors state that “However, the short-term model can be affected by compounding error if only the baseline antibody level 0 is provided.” I agree with this statement. I was wondering if this is the case for the author’s data (e.g. due to missingness) and if yes, how often is it the case?

-  For me the following formulation is unclear: “It is worth noting that the impact of the past vaccination history (before −1) is implicitly used by the model via the last antibody level −1, avoiding the need to explicitly include them, owing to the exponential modeling”. I agree that the use of the past vaccination is redundant as it is included in the last vaccination. However, it reads as if this fact is because of the exponential model - which is not clear to me.

1.4. Regression

-  The authors state that missing values are omitted for both long-term and short-term models; however, they do not justify this decision. I think it would be important to assess whether the dataset used for model training (i.e., without missing values) is a representative subsample of the original data. If the missing values follow a certain pattern, the model’s results might be biased to the characteristics of the training data and not generalizable to the whole dataset.

- It is mentioned that the logarithm of antibody titer level is used for modeling. I believe this should be incorporated into the supplementary, such that the model’s specification follows:

- It is mentioned that the dataset was divided into training and test sets only once, with a 80-20 split. However, it is not mentioned whether authors used an internal cross-validation on the training set for parameter tuning of the Explainable Boosting Machines (EBMs) model. I believe it is important to state how the EBM tuning was performed to enhance the reader's understanding on how the EBMs parameters were chosen.

- Furthermore, splitting the data only once results in having one realization for the evaluation metrics, which are random variables due to their dependence on the sample. The robustness of the results could be enhanced by including outer-cross-validation. As the authors currently split the data by 80% and 20%, an outer 5-fold cross validation might be advisable.

- The authors employ linear regression and decision trees as benchmark models. It is unclear to me why they did not consider using more complex models such as XGBoost or LightGBM as comparison models, as these are widely recognized for superior prediction accuracy over linear regression and decision trees. I believe it would be more interesting to compare the EBM to such more robust approaches to have a solid benchmark.

- Furthermore, as the authors motivate their model as being different to the exponential model, it would be interesting to see how the exponential model performs on their data.

1.5. Half-life estimation

- In this section, the authors mention that MenAfriVac as the first vaccination is the only one used for the half-life modeling. However, this decision is not further justified, so it is unclear to me whether this has a clinical reason or rather a reason from the structure of the dataset itself.

- It is stated that individuals that do not reach the immunity threshold of 128 are excluded for the half-life analysis. As this means that the half life is only valid for individuals that reach this threshold, I suggest to mention that the half life is actually a half-life conditioned on a positive response to the vaccine.

- The authors first speak about 99% Confidence Intervals and then about 95% Confidence intervals. Please clarify.

- What does it mean that they bootstrapped the samples  “and predicted their future antibody levels, incrementally adjusting the model by one day”?

1.6. Results

- The authors evaluate and compare EBM against the benchmark models using the R-squared and RMSE metrics. They mention the following: “The EBM model performed better than both comparator regression models, both in long-term and short-term approaches, with respect to both metrics”. However, the evaluation of the model’s performance using both R-square and RMSE is redundant, as the R-Square is just a monotone transformation of the MSE. Thus, I would suggest only reporting one metric if the authors do not have a good reason to report both (in that case, I still suggest to report that both metrics favor the same models by construction). The statement “with respect to both metrics” is, however, in both cases misleading.

- The authors make the following observation: “Finally, we found that using more samples resulted in the best performance, suggesting that more data might further improve our models in the future, irrespective of the study protocol”. I did not understand what is the purpose of testing the model with different amounts of observations, as it is well understood that using more available information will increase the model’s prediction accuracy.

1.7. Supplementary

- In the section “Formulation of the machine learning approach from mixed effects model”, there is a typo in “[...] while Kl depends on Ab0  , the index of K should be L and not I.

- It was not clear to me immediately whether the supplementary information on the model’s mathematical formulation refers to the long-term, short-term modeling, or a generalization of both models. I understand that is rather the latest, where depending on whether it is a long or short-term model, the term Abn will be different. However, I think that stating this explicitly and putting examples on how the formulation could change for the long and short-term model could enhance the reader’s understanding in this section.

- The authors state: “Following White et al’s model, we used the log values of the antibody level, considering its exponential dependence on time. Another alternative was to take exponent of time as a feature while keeping the antibody level as is, however, this did not work well in our preliminary experiments; hence we limited our experiments to the first approach only.”. I think it could be interesting to explain further why it was that the exponential approach did not perform well in their model. This could improve the reader’s understanding on when to use log transformation or the exponential of time for modeling antibody responses.

- Throughout the feature engineering section, there are strange characters in the text. This persists with different pdf viewers.

- In the second point on the “Feature engineering” section, the authors mention that they created features using individual vaccine doses information. Specifically, they mention: “Some examples of original features associated with each dose are the type of the vaccine (PsA-TT or Hib-TT or something else), the dose amount, the time when it was administered, etc.”. I don’t think the authors should summarize which features were available for later feature creation with the term “etc”, as it is important for the reader to understand which variables were available to later create the new variables used for the models. Thus, I would suggest listing the full original features available.

- Later on, they mention: “For the long-term model, we derived several features like the total dose of vaccines (both overall and PsA-TT only), the time of the most recent vaccine, the mean administering time of vaccines, etc.”. Again, the term “etc” should not be used, but instead the authors could refer to Table S1 which lists the features used for both long and short-term models.

1.8. General style

-  Mathematical expressions and symbols within text should be formatted appropriately, as they look like it is not a formula but rather plain text.            

2. Minor issues

2.1. Introduction

- In the sentence: “While adults get infected, children and adolescents do so at much higher rates”, a citation is needed to complement the statement.

- The term medical variables is too broad; the variables are described later in the paper but could be good to already give a brief overview in the introduction (also because it is mentioned that this is one of the main differences between the authors’ approach and White et al.).

2.2. Half-life estimation

- The sentence “For these data points, we used the initial antibody level (0) as input if it was above the threshold. If not, we used the highest recorded antibody level for the individual. Individuals with no recorded antibody levels above the threshold were excluded from the analysis” is a bit confusing. Maybe it is good to explain that if an individual does not record antibody levels above the threshold at any time point, it is excluded from the analysis.

Reviewer #2: Major comments:

• Overall this is an interesting manuscript using innovative methods to estimate duration of protection of MenAfriVac, which is an important public health and policy question that has been previously answered but can benefit from being revisited with newer methods.

More detailed comments:

Abstract:

• Is there an extra word in this sentence? “In the short-term model, we found moderately high performance (R-squared = 0.59) for out-of-training-data subjects and END even better performance (R squared = 0.83) in the long-term evaluation.”

Methods:

• In the half-life estimation methods, “If an individual had no recorded antibody levels above the threshold, the subject was excluded from the analysis.” Can you explain this a bit more? I would imagine that individuals that don’t respond to the vaccine would be important to include in calculating/estimating population half-life, as they don’t help increase population immunity. If vaccine non-responders are excluded, then the estimated half-lives only pertain to those individuals that respond to the vaccine, so might be then artificially high when thinking about population-wide immunity. Could non-responders be considered to contribute 0 time to the half-life calculations, as they spent no time above the immunity threshold?

Discussion:

• It looks like there is a missing reference in the first paragraph.

• WHO policy recommendations for the new lower-cost pentavalent conjugate vaccine are available at: Meningococcal vaccines: WHO position paper on the use of multivalent meningococcal conjugate vaccines in countries of the African meningitis belt, January 2024

• It would be interesting to also compare these estimated MenAfriVac duration results to estimates of duration of protection for other meningococcal conjugate vaccines

Figures

• Figure 4 – can the x-axis be labeled?

6. PLOS authors have the option to publish the peer review history of their article (what does this mean? ). If published, this will include your full peer review and any attached files.

**Do you want your identity to be public for this peer review?** For information about this choice, including consent withdrawal, please see our Privacy Policy .

Reviewer #1: No

Reviewer #2: **Yes: ** Heidi M. Soeters

---

## [Author Response · Author response to Decision Letter 1]

21 Oct 2024

Dear Editor(s) and Reviewers,

Thank you so much for reviewing the manuscript and providing your valuable comments to improve the quality of the manuscript. We have addressed the comments and made appropriate edits in the manuscript. Our responses are shown below (followed by "") along with the comments from the reviewers.

Reviewer #1:

The manuscript “Modeling protective meningococcal antibody responses and factors influencing antibody persistence following vaccination with MenAfriVac® using machine learning” by Nasir et al. addresses the public health challenge posed by meningococcal meningitis in the meningitis belt region in Sub-Saharan Africa. The authors argue that despite the introduction of the MenAfriVac vaccine to eliminate cases of Neisseria meningitidis serogroup A (NmA) in this region, the duration of post-vaccination immunity and the necessity of booster doses remains unclear. To address this gap, the authors propose a long and short-term model that uses demographic, medical and antibody variables to predict NmA antibody responses, as well as vaccine’s half-life. They use Explainable Boosting Machines (EBM) to model the long and short-term models and later they benchmark these results with linear regression and decision trees, where the EBM model shows better performance.

The study is well motivated by the necessity of assessing post-vaccination immunity. It also proposes two interesting approaches (long and short-term) for predicting antibody kinetics.

However, the manuscript has in my opinion some shortcomings:

1. Major issues

1.1. Introduction

- The authors state the following: “Under the current study, we developed new computational models using machine learning techniques in order to improve the utility of modeling to inform decisions on PsA-TT scheduling and dosing”. It is unclear to me whether they refer to the approach/formulation of long and short-term models or rather to the EBM model. If they refer to the former, this is not really a new development of a computational model but a rather new formulation on how to predict antibody kinetics based on long and short term responses. If the authors refer to the latter, it should be stated more clearly that they have contributed to the development of the EBM tool.

-- We do not propose EBM model. We adopted this predictive machine learning algorithm for our application of predicting antibody levels with two different formulations of the problem and suitable feature engineering for them. As suggested, we replaced ‘new computational models’ with ‘new computational frameworks’ to better reflect the contribution in the revised manuscript.

1.2. Dataset

- The exact variables which are available are unclear. Some demographic information is mentioned in the text and the authors refer to the supplementary information for the reader to check information on vaccination data. However, no such information was found in the supplementary S1. Referring to other variables as “etc” in the main text is not sufficient if an explicit list is not available in the supplementary. Table S1 contains information on variables incorporated into the long and short-term models, but as demographic information is missing from that table, it is unclear which data is really available for the authors.

-- Provided the list of variables in Table S1.

- There is no description of the cohort on the observables from the data collected (i.e., distribution of demographic variables, vaccination doses, clinical data). Descriptive statistics of the study variables will benefit the understanding of the cohort’s characteristics.

-- Provided the descriptive statistics of variables in Table S1.

1.3. Modelling

General

- The formal equations of the model in the supplementary material S1 are only referred to at the end of the short-term model description in the main text. Having the reference earlier (i.e., when describing the long-term model) could enhance clarity of the model formulation.

-- Agreed. We added a reference to the formal equations while describing the long-term model to motivate our formulation.

- The authors model each antibody task independently. This might lead to the loss of shared information between the tasks.

-- This is a completely valid point and was taken into full consideration while conceptualization of the problem formulation. However, this was not feasible to model the task as time series forecasting with shared information across readings within a single subject due to the inconsistency in the time gap between events like vaccination and antibody readings. Moreover, since the proposed model does not require more than one previous reading, it has the additional benefit of real-world applicability at the onset of a new study.

- Antibody information often comes with a lower/upper threshold of the data. Is this the case here? If yes, the authors should consider censored regression using appropriate loss functions.

-- The lower limit of quantitation for the assay was 4. Anything below that was given the value of LLOQ/2, which equals 2. There was no upper limit. While we did not use censored regression as the current study focused on antibody levels as an effect of vaccination which rarely involved the lower limits, but it would be interesting to explore this as future work, particularly if the dataset and goal of the modeling involve many values below the threshold.

Long-term modeling:

- While figure 1 a-b illustrates well which antibody readings are used for the prediction task, the readability of the text could benefit from including the information that the model only uses the baseline antibody level and no further readings are included.

-- Thank you for the suggestion! We added a phrase in the caption of the figure 1a for long-term modeling and a similar phrase for its short-term counterpart to highlight this.

- The authors claim that “A model limitation is that the estimate could be less accurate for small values of as the model is unlikely to encounter such time ranges with much frequency.” This should follow from the aforementioned sample description with descriptive statistics.

-- The descriptive statistics for the time range have been included to support the claim.

Short-term modeling:

- The authors state that “Prior studies have shown that the antibody level has an exponential relationship with time.[11] As a result, limiting the training samples to shorter ranges of time makes the model less prone to error, particularly if multiple readings from the same subject are available.” I agree that shorter time-intervals might reduce the variance of a prediction task but I do not see why this should be, relatively, higher for exponential relationships if the applied model allows for non-linear relationships.

-- Even though the model allows for the nonlinear relationship, the presence of noise leads to imperfect estimation of the parameters inherent to the model that captures this relationship. Similar levels of error in this estimation would lead to larger errors for the long-term model than the short-term one as the multiplier to the exponent is higher for longer time duration.

- The authors state that “However, the short-term model can be affected by compounding error if only the baseline antibody level 0 is provided.” I agree with this statement. I was wondering if this is the case for the author’s data (e.g. due to missingness) and if yes, how often is it the case?

-- The statement was intended to suggest a real-world test case scenario instead of reflecting the data. If applied for a new study with only the baseline reading available, the short-term model might have compounding errors to predict antibody levels at multiple points of time in the future while using the predicted level at each point as baseline for the next prediction.

-For me the following formulation is unclear: “It is worth noting that the impact of the past vaccination history (before −1) is implicitly used by the model via the last antibody level −1, avoiding the need to explicitly include them, owing to the exponential modeling”. I agree that the use of the past vaccination is redundant as it is included in the last vaccination. However, it reads as if this fact is because of the exponential model - which is not clear to me.

-- We agree that it is not necessarily due to the exponential modeling. Hence, we removed the phrase attributing the behavior to the exponential modeling in the manuscript.

1.4. Regression

- The authors state that missing values are omitted for both long-term and short-term models; however, they do not justify this decision. I think it would be important to assess whether the dataset used for model training (i.e., without missing values) is a representative subsample of the original data. If the missing values follow a certain pattern, the model’s results might be biased to the characteristics of the training data and not generalizable to the whole dataset.

-- The missing values primarily arose from subjects who did not follow up due to reasons beyond the scope of the study and quality/measurement issues in the titer assay for antibody readings. As none of these reasons may indicate any characteristics of immunogenicity, we did not further analyze the missingness of the data.

- It is mentioned that the logarithm of antibody titer level is used for modeling. I believe this should be incorporated into the supplementary, such that the model’s specification follows:

-- Agreed. We added a sentence to the supplementary mentioning this.

- It is mentioned that the dataset was divided into training and test sets only once, with a 80-20 split. However, it is not mentioned whether authors used an internal cross-validation on the training set for parameter tuning of the Explainable Boosting Machines (EBMs) model. I believe it is important to state how the EBM tuning was performed to enhance the reader's understanding on how the EBMs parameters were chosen.

-- We indeed tuned the hyperparameters of the EBM by 5-fold internal cross-validation for each instance of model training. Once the optimal hyperparameters were obtained, we trained the model on the entire training set using those parameters. This information has been added to the manuscript as follows: “ We conducted a 5-fold cross validation over the training data to obtain the optimal parameters for EBM and used them for finally training the model on the entire training set.”

-Furthermore, splitting the data only once results in having one realization for the evaluation metrics, which are random variables due to their dependence on the sample. The robustness of the results could be enhanced by including outer-cross-validation. As the authors currently split the data by 80% and 20%, an outer 5-fold cross validation might be advisable.

-- We completely agree that it would have been more robust to perform outer cross-validation with different training and test splits of the dataset. However, in the scope of the current study, we chose not to do so as it would be computationally expensive, especially due to having multiple proposed and baseline models. Nonetheless, we conducted a bootstrapping experiment with half-life estimation to evaluate its robustness in terms of 95% confidence intervals.

- The authors employ linear regression and decision trees as benchmark models. It is unclear to me why they did not consider using more complex models such as XGBoost or LightGBM as comparison models, as these are widely recognized for superior prediction accuracy over linear regression and decision trees. I believe it would be more interesting to compare the EBM to such more robust approaches to have a solid benchmark.

-- Thank you for the suggestion. We conducted new experiments with XGBoost and LightGBM as baselines and reported their performance in Table 2. We also added a few sentences in the Methods section (under Regression) and the Results section referring to these baseline models.

- Furthermore, as the authors motivate their model as being different to the exponential model, it would be interesting to see how the exponential model performs on their data.

-- While it would have been really interesting to see how the proposed model’s performance with White et al’s exponential model, there is a fundamental difference between these models in how each data point is constructed. Each exponential model considers a single subject as one sample and predicts the future antibody level using multiple past values, whereas in our framework one sample constitutes only one past reading to predict the future level. So, it would be unfair to compare the performances across these models.

1.5. Half-life estimation

- In this section, the authors mention that MenAfriVac as the first vaccination is the only one used for the half-life modeling. However, this decision is not further justified, so it is unclear to me whether this has a clinical reason or rather a reason from the structure of the dataset itself.

-- In the scope of MenAfriVac vaccine studies, the half-life of a vaccine is defined with respect to a single dose without any residual effect of a previous dose. This is why the half-life estimation analysis was limited to antibody levels following the first vaccination only before the next dose was administered.

- It is stated that individuals that do not reach the immunity threshold of 128 are excluded for the half-life analysis. As this means that the half life is only valid for individuals that reach this threshold, I suggest to mention that the half life is actually a half-life conditioned on a positive response to the vaccine.

-- This is a great suggestion. We added the following to clarify this in the manuscript: “It should be noted that our definition of half-life is conditioned on a positive response to the vaccine. Hence if an individual had no recorded antibody levels above the threshold, the subject was excluded from the analysis.”

- The authors first speak about 99% Confidence Intervals and then about 95% Confidence intervals. Please clarify.

-- The mention of the 99% confidence interval was due to a typo. We have fixed it in the revised manuscript and thank the reviewer for finding it.

- What does it mean that they bootstrapped the samples “and predicted their future antibody levels, incrementally adjusting the model by one day”?

-- We recognized that the phrase might be vague, and rephrased and clarified the statement as “predicted their future antibody levels one day at a time, incrementally applying the model for the next day using the previous day’s antibody level as an input.”

1.6. Results

- The authors evaluate and compare EBM against the benchmark models using the R-squared and RMSE metrics. They mention the following: “The EBM model performed better than both comparator regression models, both in long-term and short-term approaches, with respect to both metrics”. However, the evaluation of the model’s performance using both R-square and RMSE is redundant, as the R-Square is just a monotone transformation of the MSE. Thus, I would suggest only reporting one metric if the authors do not have a good reason to report both (in that case, I still suggest to report that both metrics favor the same models by construction). The statement “with respect to both metrics” is, however, in both cases misleading.

-- We fully agree with the point raised about the monotonic relationship between R-squared and RMSE. We reported both metrics as we wanted to capture the numeric range or errors (through RMSE) as well as the standardized goodness of fit (through R-squared). We now replaced the phrase “with respect to both metrics” by “as reflected in the reported metrics”. We also added a sentence in the methodology to enunciate that both metrics favor the same models by construction and provide the rationale behind using both.

- The authors make the following observation: “Finally, we found that using more samples resulted in the best performance, suggesting that more data might further improve our models in the future, irrespective of the study protocol”. I did not und

---

## [Decision Letter · Decision Letter 1]

13 Feb 2025

PONE-D-24-05900R1Modeling protective meningococcal antibody responses and factors influencing antibody persistence following vaccination with MenAfriVac® using machine learningPLOS ONE

Dear Dr. Nasir,

Thank you for submitting your manuscript to PLOS ONE. After careful consideration, we feel that it has merit but does not fully meet PLOS ONE’s publication criteria as it currently stands. Therefore, we invite you to submit a revised version of the manuscript that addresses the points raised during the review process. Please submit your revised manuscript by Mar 30 2025 11:59PM. If you will need more time than this to complete your revisions, please reply to this message or contact the journal office at plosone@plos.org . Please include the following items when submitting your revised manuscript:

We look forward to receiving your revised manuscript.

Kind regards,

Oyelola A. Adegboye, PhD

Academic Editor

PLOS ONE

Journal Requirements:

Reviewers' comments:

Reviewer's Responses to Questions

**Comments to the Author**

1. If the authors have adequately addressed your comments raised in a previous round of review and you feel that this manuscript is now acceptable for publication, you may indicate that here to bypass the “Comments to the Author” section, enter your conflict of interest statement in the “Confidential to Editor” section, and submit your "Accept" recommendation.

Reviewer #1: All comments have been addressed

Reviewer #3: (No Response)

2. Is the manuscript technically sound, and do the data support the conclusions?

Reviewer #1: Yes

Reviewer #3: (No Response)

3. Has the statistical analysis been performed appropriately and rigorously? 

Reviewer #1: Yes

Reviewer #3: (No Response)

4. Have the authors made all data underlying the findings in their manuscript fully available?

Reviewer #1: No

Reviewer #3: (No Response)

5. Is the manuscript presented in an intelligible fashion and written in standard English?

Reviewer #1: Yes

Reviewer #3: Yes

6. Review Comments to the Author

Reviewer #1: I appreciate the authors' detailed responses to my inquiries. Their revisions have effectively addressed all of my concerns and significantly improved the manuscript. I have only two minor suggestions remaining; once these are addressed, I believe the manuscript will be ready for publication.

Minor Points

1)

Original Comment

Antibody information often comes with a lower/upper threshold of the data. Is this the case here? If yes, the authors should consider censored regression using appropriate loss functions.

Author’s Response

The lower limit of quantitation for the assay was 4. Anything below that was given the value of LLOQ/2, which equals 2. There was no upper limit. While we did not use censored regression as the current study focused on antibody levels as an effect of vaccination which rarely involved the lower limits, but it would be interesting to explore this as future work, particularly if the dataset and goal of the modeling involve many values below the threshold.

New Comment:

I agree with the authors that if censoring is limited, then using methods for censoring might not be necessary. However, I suggest to discuss this briefly in the paper and to mention the fraction of censored observations.

2)

Original Comment

The authors state that “However, the short-term model can be affected by compounding error if only the baseline antibody level 0 is provided.” I agree with this statement. I was wondering if this is the case for the author’s data (e.g. due to missingness) and if yes, how often is it the case?

Author’s Response

The statement was intended to suggest a real-world test case scenario instead of reflecting the data. If applied for a new study with only the baseline reading available, the short-term model might have compounding errors to predict antibody levels at multiple points of time in the future while using the predicted level at each point as baseline for the next prediction.

New Comment:

I suggest to include this reasoning in order to avoid confusions.

Reviewer #3: The manuscript “Modeling protective meningococcal antibody responses and factors influencing antibody persistence following vaccination with MenAfriVac® using machine learning” explores the modeling of protective meningococcal antibody responses and factors influencing antibody persistence following PsA-TT (MenAfriVac) vaccination. The study employs machine learning techniques, specifically Explainable Boosting Machines (EBM), to develop both short-term and long-term predictive models. These models incorporate demographic, clinical, and serological data to estimate antibody kinetics and vaccine half-life. The work is well motivated, potentially being relevant in the context of optimizing vaccination strategies, well-structured and generally clear; however, I have some concerns and suggestions that, if addressed, could enhance the clarity, reproducibility, and overall transparency of the study.

Major concern:

- Authors should provide a more detailed description of the pre-processing steps applied to both predictive and outcome variables? While some information is provided in Supplementary 1, a more explicit explanation in the main text could improve clarity for the reader. Specifically:

• Was batch correction applied to the outcome variable, given that measurements were obtained from different studies under potentially varying experimental conditions? If not, please clarify the choice

• Were predictive variables subjected to normalization, decorrelation or other pre-processing techniques? Since the performance of some models used for comparison may be sensitive to these pre-processing choices, further details would be beneficial.

Minor concerns:

• The manuscript compares the model proposed by White et al with Linear Regression, Regression Trees, LightGBM, and XGBoost. Could the authors elaborate on the rationale for selecting these specific models? For example, were other ensemble or deep learning methods considered? A brief justification of these choices would provide a better context.

• In the "Half-life estimation" section, the authors define a protective antibody threshold of 128 rSBA. Could they provide a reference or a more detailed explanation supporting this threshold? A brief discussion on why this value is considered appropriate would strengthen the validity of this assumption.

• The "Model Interpretation" section presents feature importance rankings in Figures 4a and 4b, limited to the top 15 features. Would it be possible to provide a table or supplementary plot including all feature importance values? This would enhance transparency and allow readers to assess the relative contributions of less influential variables.

• In the "Discussion" section, the phrase "we describe the development and evaluation of a machine learning algorithm to characterize..." might be somewhat misleading, as the original model was developed by White et al. A more precise wording, emphasizing the application and evaluation rather than the development of the model itself, would be more appropriate.

• Supplementary 1: The subscripts a, l, and s are not clearly explained. A brief clarification would enhance readability.

• Tables S1 and S2: Do these tables share the same legend? Please write a separate legend for each Table.

• In the "Regression" section, the manuscript states that the long-term model includes 39 variables, while the short-term model includes 14. However, these numbers do not align with those in Table S2. Could the authors clarify whether this discrepancy arises from interaction terms, dummy encoding of categorical variables or something else?

7. PLOS authors have the option to publish the peer review history of their article (what does this mean? ). If published, this will include your full peer review and any attached files.

**Do you want your identity to be public for this peer review?** For information about this choice, including consent withdrawal, please see our Privacy Policy .

Reviewer #1: No

Reviewer #3: No

---

## [Author Response · Author response to Decision Letter 2]

28 Mar 2025

Dear Editor(s) and Reviewers,

Thank you so much for reviewing the manuscript and providing your valuable comments to improve the quality of the manuscript. We have addressed the comments and made appropriate edits in the manuscript. Our responses are shown below along with the comments from the reviewers (in red font).

Reviewer #1: I appreciate the authors' detailed responses to my inquiries. Their revisions have effectively addressed all of my concerns and significantly improved the manuscript. I have only two minor suggestions remaining; once these are addressed, I believe the manuscript will be ready for publication.

--Thank you so much for your feedback. We really appreciate your rigorous and insightful suggestions which we believe have improved the quality of the manuscript.

Minor Points

1)

Original Comment

Antibody information often comes with a lower/upper threshold of the data. Is this the case here? If yes, the authors should consider censored regression using appropriate loss functions.

Author’s Response

The lower limit of quantitation for the assay was 4. Anything below that was given the value of LLOQ/2, which equals 2. There was no upper limit. While we did not use censored regression as the current study focused on antibody levels as an effect of vaccination which rarely involved the lower limits, but it would be interesting to explore this as future work, particularly if the dataset and goal of the modeling involve many values below the threshold.

New Comment:

I agree with the authors that if censoring is limited, then using methods for censoring might not be necessary. However, I suggest to discuss this briefly in the paper and to mention the fraction of censored observations.

--We added the following in the Dataset section: “The lower limit of quantitation (LLOQ) for the assay was 4, and values below this threshold were assigned LLOQ/2 (i.e., 2). In our dataset, less than 1% of observations were below the LLOQ.”

We also mentioned in the “Conclusion and Future Work” section: “Given the limited number of censored values, we did not apply censored regression techniques. However, future studies with a higher fraction of censored values may benefit from such approaches to account for measurement limitations more rigorously.”

2)

Original Comment

The authors state that “However, the short-term model can be affected by compounding error if only the baseline antibody level 0 is provided.” I agree with this statement. I was wondering if this is the case for the author’s data (e.g. due to missingness) and if yes, how often is it the case?

Author’s Response

The statement was intended to suggest a real-world test case scenario instead of reflecting the data. If applied for a new study with only the baseline reading available, the short-term model might have compounding errors to predict antibody levels at multiple points of time in the future while using the predicted level at each point as baseline for the next prediction.

New Comment:

I suggest to include this reasoning in order to avoid confusions.

--We edited our initial statement to include this reasoning.

Reviewer #3: The manuscript “Modeling protective meningococcal antibody responses and factors influencing antibody persistence following vaccination with MenAfriVac® using machine learning” explores the modeling of protective meningococcal antibody responses and factors influencing antibody persistence following PsA-TT (MenAfriVac) vaccination. The study employs machine learning techniques, specifically Explainable Boosting Machines (EBM), to develop both short-term and long-term predictive models. These models incorporate demographic, clinical, and serological data to estimate antibody kinetics and vaccine half-life. The work is well motivated, potentially being relevant in the context of optimizing vaccination strategies, well-structured and generally clear; however, I have some concerns and suggestions that, if addressed, could enhance the clarity, reproducibility, and overall transparency of the study.

--We really appreciate your constructive comments. We have addressed them below and updated the manuscript accordingly.

Major concern:

Authors should provide a more detailed description of the pre-processing steps applied to both predictive and outcome variables? While some information is provided in Supplementary 1, a more explicit explanation in the main text could improve clarity for the reader.

Specifically:

• Was batch correction applied to the outcome variable, given that measurements were obtained from different studies under potentially varying experimental conditions? If not, please clarify the choice

--We added the following in the main text (in the “Regression” subsection): “Due to variations in the design of the original constituent studies—such as differences in the number of vaccinations and the timing of vaccine administration—we derived summary variables for the long-term model. For instance, we used the mean time of vaccination rather than individual vaccination time points.”

• Were predictive variables subjected to normalization, decorrelation or other pre-processing techniques? Since the performance of some models used for comparison may be sensitive to these pre-processing choices, further details would be beneficial.

--Thank you for the suggestion of highlighting this. We indeed performed some preprocessing step for linear regression, only one of the baselines. We included the following in the text: “For linear regression, the predictor variables were normalized, and the categorical variables were encoded with one-hot encoding. EBM and the other three baseline models did not require any such preprocessing due to their scale-agnostic nature.”

Minor concerns:

• The manuscript compares the model proposed by White et al with Linear Regression, Regression Trees, LightGBM, and XGBoost. Could the authors elaborate on the rationale for selecting these specific models? For example, were other ensemble or deep learning methods considered? A brief justification of these choices would provide a better context.

--Thank you for your thoughtful comment. We selected Linear Regression, Regression Trees, LightGBM, and XGBoost for comparison primarily due to their balance between predictive performance and interpretability. Given that our goal was to evaluate and compare model interpretability—particularly with the Explainable Boosting Machine (EBM)—we prioritized models that offer transparent or explainable mechanisms.

Linear Regression and Regression Trees are inherently interpretable and serve as useful baselines. LightGBM and XGBoost, while more complex, provide feature importance metrics and can be interpreted. Although we considered other ensemble and deep learning methods, we excluded them due to their limited interpretability in the context of this study’s objectives. We have clarified this rationale in the revised manuscript (subsection “Regression”).

• In the "Half-life estimation" section, the authors define a protective antibody threshold of 128 rSBA. Could they provide a reference or a more detailed explanation supporting this threshold? A brief discussion on why this value is considered appropriate would strengthen the validity of this assumption.

--We thank the reviewer for this valuable comment. The rSBA threshold of ≥128 was chosen based on prior studies that used rSBA as a correlate of protection in the context of meningococcal conjugate vaccines. Specifically, Andrews et al. (2003) validated this threshold by comparing rSBA titers to protective levels measured by hSBA and found that an rSBA titer of ≥128 reliably predicts protection, despite being more conservative than hSBA-based thresholds. While rSBA titers in the 8–64 range were found to be equivocal, titers ≥128 consistently indicated protective immunity. We have added a citation and brief explanation to the manuscript to clarify this choice.

• The "Model Interpretation" section presents feature importance rankings in Figures 4a and 4b, limited to the top 15 features. Would it be possible to provide a table or supplementary plot including all feature importance values? This would enhance transparency and allow readers to assess the relative contributions of less influential variables.

--Thank you for the insightful suggestion. While we agree having the ability to compare with all feature importance values would have been interesting, the issue of doing so lies in the fact that there are many pairwise interaction features (13*12/2=78 for long-term model) along with the original variables make feature set fairly large (78+13=91 features in total) to meaningfully make such comparison across all features. Also, the scores for relative less important features appeared tapering off (below 0.1) beyond top 10-15 and due to linear nature of the model, their relative importances are practically insignificant. As an example, from the modeling perspective, it does not matter much whether a less important feature’s importance score is 0.01 or 0.05, as it is not significant compared to top feature importance values either way. Hence, we restricted the visualization to 15 features to keep it concise and informative.

• In the "Discussion" section, the phrase "we describe the development and evaluation of a machine learning algorithm to characterize..." might be somewhat misleading, as the original model was developed by White et al. A more precise wording, emphasizing the application and evaluation rather than the development of the model itself, would be more appropriate.

--While the statistical model was developed by While et al, we did not use it directly, we just derived the predictor variables and took inspiration from original statistical model to formulate a regression problem (in two alternative frameworks- long-term and short-term model) and used machine-learning based algorithms to solve the regression. This is in contrast with White et al’s model, which was aimed at fitting the data directly and extrapolation. Despite these fundamental differences, we do agree that rephrasing the sentence would be more appropriate. We made the following revision:” In this article, we describe the formulation and evaluation of a machine learning-based framework to characterize and predict the kinetics of serum antibody response to meningococcal serogroup A vaccination.”

• Supplementary 1: The subscripts a, l, and s are not clearly explained. A brief clarification would enhance readability.

--Thank you for the valuable suggestion. We added the clarification: “where 〖Ab〗_0 is the initial antibody level and r_l, r_s, and r_a are decay rates of different types of antibody secreting cells (long-lived, short-lived, and all combined) responsible for vaccine-induced immunity. “

• Tables S1 and S2: Do these tables share the same legend? Please write a separate legend for each Table.

--Thanks for catching this. The legends in Table S2 were dropped since they were not used in the text.

• In the "Regression" section, the manuscript states that the long-term model includes 39 variables, while the short-term model includes 14. However, these numbers do not align with those in Table S2. Could the authors clarify whether this discrepancy arises from interaction terms, dummy encoding of categorical variables or something else?

--Yes, this was indeed due to dummy (one-hot) encoding of the categorical variables. To avoid confusion, we now updated the manuscript to report the number of original variables and later mention the one-hot encoding as preprocessing step.

We appreciate your consideration of the revised manuscript for possible publication in PLOS One and look forward to hearing back from you.

Best regards,

Md Nasir

Dr. Md Nasir for the authors

Senior Research Scientist

AI for Good Research Lab, Microsoft

Email: mdnasir@microsoft.com

Postal address: 1 Microsoft Way, Redmond, WA, 98052, USA.

---

## [Decision Letter · Decision Letter 2]

8 Apr 2025

Modeling protective meningococcal antibody responses and factors influencing antibody persistence following vaccination with MenAfriVac® using machine learning

PONE-D-24-05900R2

Dear Dr. Weeks,

We’re pleased to inform you that your manuscript has been judged scientifically suitable for publication and will be formally accepted for publication once it meets all outstanding technical requirements.

Kind regards,

Oyelola A. Adegboye, PhD

Academic Editor

PLOS ONE

Additional Editor Comments (optional):

Reviewers' comments:

Reviewer's Responses to Questions

**Comments to the Author**

1. If the authors have adequately addressed your comments raised in a previous round of review and you feel that this manuscript is now acceptable for publication, you may indicate that here to bypass the “Comments to the Author” section, enter your conflict of interest statement in the “Confidential to Editor” section, and submit your "Accept" recommendation.

Reviewer #1: All comments have been addressed

Reviewer #3: All comments have been addressed

2. Is the manuscript technically sound, and do the data support the conclusions?

Reviewer #1: Yes

Reviewer #3: Yes

3. Has the statistical analysis been performed appropriately and rigorously? 

Reviewer #1: Yes

Reviewer #3: Yes

4. Have the authors made all data underlying the findings in their manuscript fully available?

Reviewer #1: Yes

Reviewer #3: Yes

5. Is the manuscript presented in an intelligible fashion and written in standard English?

Reviewer #1: Yes

Reviewer #3: Yes

6. Review Comments to the Author

Reviewer #1: The authors have addressed my remaining comments and the manuscript is in my opinion ready for publication.

Reviewer #3: (No Response)

7. PLOS authors have the option to publish the peer review history of their article (what does this mean? ). If published, this will include your full peer review and any attached files.

**Do you want your identity to be public for this peer review?** For information about this choice, including consent withdrawal, please see our Privacy Policy .

Reviewer #1: No

Reviewer #3: No

---

## [Editor Report · Acceptance letter]

PONE-D-24-05900R2

PLOS ONE

Dear Dr. Weeks,

I'm pleased to inform you that your manuscript has been deemed suitable for publication in PLOS ONE. Congratulations! Your manuscript is now being handed over to our production team.

Kind regards,

on behalf of

Assoc Prof Oyelola A. Adegboye

Academic Editor

PLOS ONE